# Narrow-Linewidth Single-Frequency Ytterbium Laser Based on a New Composite Yb³⁺-Doped Fiber

**Maksim Yu. Koptev** [1], **Olga N. Egorova** [2], **Oleg I. Medvedkov** [2], **Sergey L. Semjonov** [2], **Boris I. Galagan** [2], **Sergey E. Sverchkov** [2], **Boris I. Denker** [2], **Alexander E. Zapryalov** [1] **and Arkady V. Kim** [1,*]

[1] Institute of Applied Physics of the Russian Academy of Sciences, 46 Ulyanov Str., 603950 Nizhny Novgorod, Russia

[2] Prokhorov General Physics Institute of the Russian Academy of Sciences, 38 Vavilov Str., 119333 Moscow, Russia

\* Correspondence: arkady.kim@gmail.com

**Abstract:** Fiber single-frequency lasers are currently being actively developed, primarily due to the growing number of applications that require compact and reliable narrow-band sources. However, the most developed single-frequency fiber lasers based on phosphate fibers have the disadvantages of low mechanical strength of both the phosphate fibers themselves and their splices. In this paper we demonstrate a single-frequency laser based on a new composite Yb³⁺-doped active fiber. The core of this fiber is made of phosphate glass with a high concentration of ytterbium ions and its cladding is made of standard silica glass. This structure ensures a higher splicing strength of the fiber compared to the phosphate fibers and provides high resistance to atmospheric moisture. Despite the multimode structure of this fiber, we achieved stable single-frequency lasing with an average power of 10 mW and a spectral contrast of more than 60 dB in the scheme with a short (1.1 cm) cavity formed by two fiber Bragg gratings. We believe that further optimization of this fiber will make it possible to create powerful and reliable single-frequency lasers in the one-micron wavelength range.

**Keywords:** composite phosphate fiber; fiber highly doped with ytterbium; short-cavity fiber laser; single-frequency fiber laser

## 1. Introduction

Narrow-band lasers operating in a single longitudinal mode (single-frequency lasers) are widely used in high-resolution sensors, spectrometers, and WDM telecommunication, and for the detection of gravitational waves and the creation of multikilowatt laser systems based on coherent combining of laser beams [1–3]. Such a wide range of single-frequency laser applications is due to their characteristics, including a narrow spectral band (tens of kilohertz), a low level of optical noise, and a high stability of output parameters. Single-frequency fiber lasers are being actively developed, primarily due to their compactness compared to solid-state counterparts and better output characteristics compared to semiconductor lasers. There are two main types of such lasers: lasers with a long (several meters), usually ring, cavity, and lasers with a short linear cavity—distributed feedback (DFB) and distributed Bragg reflector (DBR) lasers.

The first class includes lasers with a ring cavity and spectral filtering using Fabry–Perot standards [4,5], lasers with a filter based on a saturable absorber [6–8], and lasers with ring sub-cavities [9,10]. Lasers of this class can be tunable over a wide range of wavelengths, but they tend to have a large number of components and are difficult to fabricate in most cases. Their main disadvantage is a small longitudinal mode pitch (typically tens of MHz), as a result of which they are highly susceptible to longitudinal mode hopping. So, to ensure stable operation, special rather complex techniques, such as piezoelectric cavity length stabilization, are usually used in lasers of this type [9].

The second class includes fiber lasers with a short linear cavity formed by Bragg gratings, which can be either external or written directly in the active fiber. Such lasers are compact, stable in operation, and easy to manufacture. However, to provide the needed mode selection executed by fiber Bragg gratings, a small (several centimeters) length of the active fiber in the cavity is required. To achieve the gain demanded for light generation in such lasers, highly doped active fibers with a high linear gain (1–5 dB/cm) are used. Since standard silica glasses cannot efficiently dissolve large concentrations of active ions, most single-frequency short-cavity fiber lasers are based on active phosphate glass fibers [11–13]. The use of active phosphate fibers makes it possible to create single-frequency lasers in the one-micron range with an active fiber length as short as 8 mm [14]. However, the integration of such fibers into all-fiber systems is significantly hampered by the difference in the physical properties of standard silica-based fibers and phosphate fibers. Firstly, phosphate glasses usually have lower glass transition temperatures (400–600 °C, compared to 1000–1200 °C for silica) and lower softening temperatures (500–700 °C versus 1500–1600 °C), which leads to low mechanical strength of the splices between phosphate and silica glass fibers [15,16]. Secondly, phosphate glass fibers have a lower tensile strength (3–5 GPa versus 10–14 GPa) compared to silica fibers, which is also significantly reduced (up to 35%) after exposing such fibers to air [17].

Thus, a special technique is needed for the creation of a reliable and weather-resistant laser based on phosphate fibers. A composite fiber design combining the advantages of silica and phosphate fibers was proposed in [18]. The core of such a fiber is made of highly doped phosphate glass, and the cladding is made of standard silica glass. This design allows strong splices to be obtained between composite and standard silica fibers, while providing a high linear gain. The silica cladding also perfectly protects the phosphate glass core from atmospheric moisture. However, due to the large difference in the refractive indices of phosphate and silica glasses, such fibers, even with a small core diameter (4–5 μm), are multimodal in the 1 μm wavelength range, which makes it impossible to record Bragg gratings both in the active fiber and in compatible passive fibers. Nevertheless, the single-mode operation of such an optical fiber can be ensured by splicing it with standard single-mode silica fibers. This is possible, since when radiation is coaxially coupled from a single-mode fiber into a multimode one, the fundamental mode is predominantly excited in the latter. Therefore, if the laser cavity is formed by Bragg gratings recorded in a single-mode fiber, single-mode generation is possible in such a laser, despite the multimode nature of the active fiber. With the use of this method, it was demonstrated that it is possible to create a single-mode laser based on a hybrid fiber with an efficiency of more than 70 percent with an active fiber length of 48 mm [19]. We note that a single-frequency fiber laser on a similar composite fiber was reported earlier in the 1.5 μm range [20], but it was made on a single-mode fiber, which made it possible to write the cavity directly in the active fiber. A single-frequency laser based on a few-mode composite fiber in the one-micron range is demonstrated by us for the first time.

In this paper, we propose a single-frequency ytterbium laser generating stable radiation with a central wavelength of 1030 nm and an average power of 10 mW based on a composite ytterbium fiber with a core made of highly doped phosphate glass and a cladding of silica glass [19]. Even though the active fiber is multimodal, splicing it with fiber Bragg gratings written in standard single-mode silica fibers makes it possible to create a laser operating in a single-frequency regime.

## 2. Materials and Methods

### 2.1. Composite Ytterbium-Doped Fiber

To create a single-frequency fiber laser, we used a new composite ytterbium-doped fiber as an active medium. The core of this fiber was made of phosphate glass containing 65 mol% $P_2O_5$, 7 mol% $Al_2O_3$, 12 mol% $B_2O_3$, 9 mol% $Li_2O$, and 7 mol% $La_2O_3$. The concentration of ytterbium ions was $5.0 \times 10^{20}$ cm$^{-3}$. An active fiber preform was fabricated

using the "rod in tube" technology by inserting a phosphate glass core into a silica tube and further consolidating the preform at a high (2000 °C) temperature.

An optical fiber with a cladding diameter of 80 μm and a core diameter of 4.8 μm was drawn from the resulting preform. The cross section of the fiber, and the measured loss spectrum, are shown in Figure 1. The numerical aperture of the resulting fiber was 0.32, and the small-signal pump absorption at a 976 nm wavelength was 2.5 dB/mm. The process of creating this fiber is described in more detail in [19]. Due to the large difference in the refractive indices of phosphate and silica glasses, this fiber is multimode, but by splicing it with standard single-mode fibers, it is possible to avoid excitation of higher modes, especially when working with short sections of the active fiber.

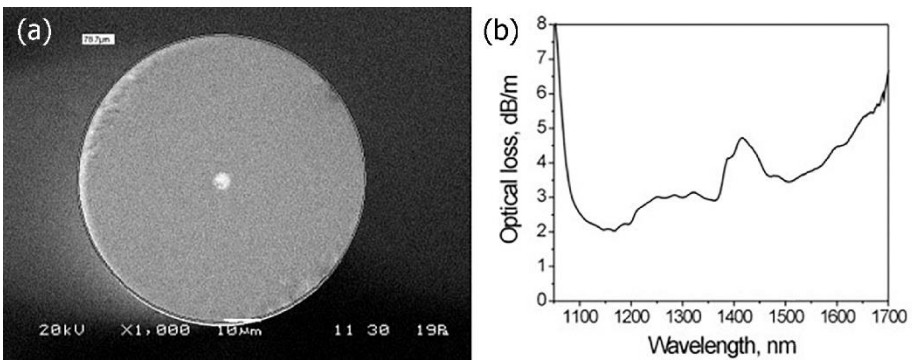

**Figure 1.** (**a**) Cross section of the Yb-doped composite fiber, (**b**) optical loss spectrum measured in the composite fiber. Reprinted with permission from [19] © The Optical Society.

### 2.2. Experimental Setup

The laser cavity was formed by two narrow-band fiber Bragg gratings, which were written in a Nufern 1060-XP single-mode fiber. The reflection coefficient of the high reflectance grating (HR-FBG) was 99.5%, its length was 5.5 mm, and the half-width of the reflection spectrum was 75 pm. The reflection coefficient of the output coupling grating (OC-FBG) was 95%, its length was 10 mm, and the width of the reflection spectrum was 35 pm. The Bragg gratings were fabricated by exposing the fiber to an ultraviolet laser through a phase mask. The reflection coefficient of the output grating was chosen to be large enough to compensate for rather large splicing losses between active and passive fibers, caused by a significant difference in their optical properties. Splicing of the active fiber with fiber Bragg gratings was carried out on a Fujikura FSM-100P+ fusion splicer, and precision cleavage was undertaken on a Fujikura CT-100 fiber cleaver. We used a standard mode for splicing optical fibers with a cladding diameter of 80 μm to splice active and passive fibers.

To estimate the losses in the obtained cavities, we used a continuous wave laser at a non-resonant wavelength of 1560 nm, whose radiation passed through the cavity. The loss in two splices was about 2.3 dB and changed insignificantly as the length of the active fiber was changed. We did not observe lasing at an active fiber length of less than 1 cm, and at a length of 1 cm, the lasing possibility strongly depended on the quality of splicing of active and passive fibers; therefore, a 1.1 cm long piece of phosphate fiber was used to build a short cavity laser. More detailed results of active fiber length optimization are given below. The resulting cavity was mounted on a glass substrate and coated with a soft polymer for better thermal conductivity and protection from external influences.

The total length of the cavity, considering the effective length of the Bragg gratings, was 14.1 mm. The effective length of the Bragg gratings was calculated in accordance with the formulas given in [21]. The longitudinal mode pitch for such a cavity is 7.3 GHz, which corresponds to a wavelength pitch of 25.8 pm. Since the Bragg gratings have some mismatch at the center wavelengths, the output grating was thermally stabilized on a Peltier

device, which was connected to a high-precision PID (proportional integral derivative) temperature controller.

The high reflecting grating was located on a radiator which had room temperature (25 °C). Since the longitudinal mode pitch was smaller than the spectral width of the output grating (35 pm), at a nonoptimal setting of the output grating temperature, lasing was observed in two longitudinal modes which appeared as two peaks in the optical spectrum. However, the operation of such a laser in the single-frequency regime is possible if one of the longitudinal modes coincides with the reflection peak of the output grating [22–24]. This condition can be easily achieved by adjusting the temperature of the OC-FBG.

The experimental setup is shown in Figure 2. The laser was backward pumped using a single-mode laser diode with a maximum output power of 400 mW spliced to a WDM-coupler. To eliminate the influence of back reflections on the generation of the laser, a Faraday isolator was placed at its output. Laser generation started from a pump power of 10 mW. The temperature of the output grating was tuned to achieve the maximum output power of the laser, which corresponds to the matching of the reflection wavelengths of the two Bragg gratings forming the cavity, as well as to match one of the longitudinal modes of the cavity with the reflection spectrum of the gratings. Its optimal temperature varied depending on the pumping power, while the room temperature (HR-FBG temperature) was 25 degrees. The longitudinal modes of the laser were monitored using a Thorlabs SA200-8B scanning Fabry–Perot interferometer with a free spectral range of 1.5 GHz and a spectral resolution of 7.5 MHz. To improve the measurement accuracy, the collimated laser beam was focused inside the interferometer cavity using a convex lens with a focal length of 250 mm. Frequency sweeping and output signal processing were performed by a Thorlabs SA201 control unit. Both the pump power and the laser output power were measured with a Thorlabs S145C integrating sphere.

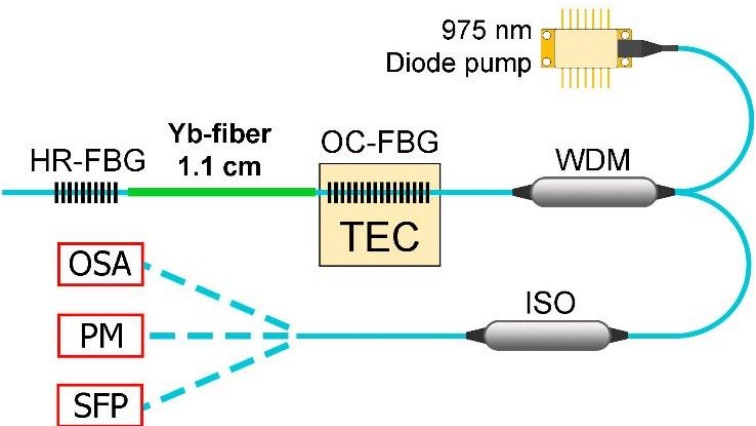

**Figure 2.** Experimental setup: HR-FBG—high reflection fiber Bragg grating, OC-FBG—output coupling fiber Bragg grating, TEC—thermoelectric cooler, WDM—wavelength division multiplexer, ISO—Faraday isolator, OSA—optical spectrum analyzer, PM—power meter, SFP—scanning Fabry–Perot interferometer.

## 3. Results

### 3.1. Optimization of the Active Fiber Length

To optimize the splicing parameters, in addition to the required length of the active fiber, a number of experiments was carried out in which the cavity length was left sufficiently large (~80 cm), and the length of the active fiber varied in the range of 1–3 cm.

As a result, the splicing current was reduced by 20% compared to the one originally set in the splicing program. Moreover, in order to avoid damage to the core of the active fiber, the fiber cleaning current was halved. Since the active fiber is multimodal, before splicing, the fibers were manually aligned along the core. Figure 3 shows the dependence of the laser output power (in the scheme with a long cavity) on the pump power for three

different active fiber lengths. With a composite fiber length of less than 1 cm, lasing was not observed, so in order to achieve stable single-frequency lasing, we chose a piece of an active fiber close to 1 cm to build a laser.

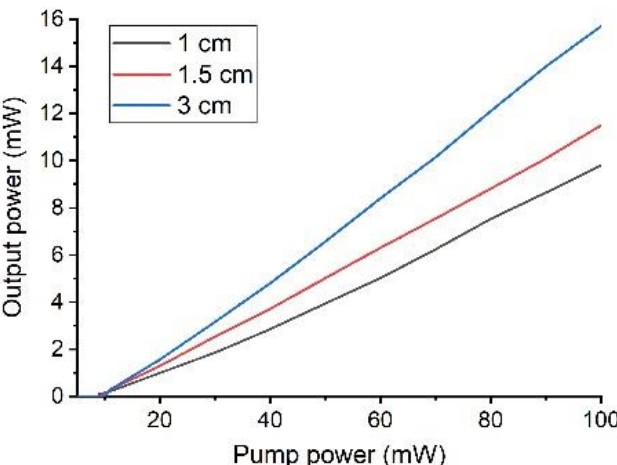

**Figure 3.** Laser output power versus pump power for three different lengths of the active fiber in a scheme with a long cavity.

### 3.2. Single-Frequency Regime

An oscillogram from a photodetector located at the output of the Fabry–Perot interferometer at a laser output power of 10 mW is shown in Figure 4. The distance between the peaks (black line in Figure 4) corresponds to the free spectral range of the interferometer, which indicates the single-frequency regime of the laser.

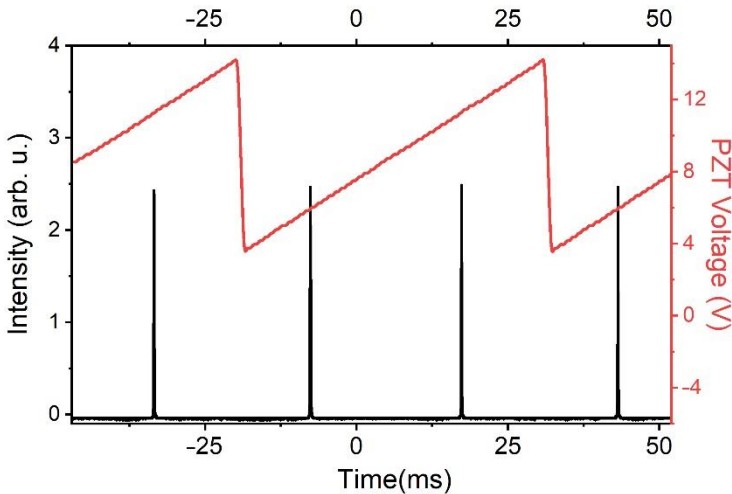

**Figure 4.** The signal at the output of the photodetector of the scanning Fabry–Perot interferometer (black line) and the scanning voltage of its piezo actuator (red line).

By analyzing the width of the peaks in Figure 4, one can estimate the width of the laser line using a simple formula $\Delta \nu_{FWHM} = FSR \cdot \Delta t_{pulse} / \Delta t_{FSR}$, where $\Delta \nu_{FWHM}$ is the estimated linewidth of the laser, FSR is the free spectral range of the interferometer, $\Delta t_{pulse}$ is the duration of the peak in Figure 4, and $\Delta t_{FSR}$ is the time between the pulses, which corresponds to the achievement of the FSR of the interferometer during scanning (1.5 GHz). So, we can say that the line width of the created laser does not exceed 7.5 MHz (the resolution of the SA200 interferometer), but we believe that the line width is actually much smaller.

### 3.3. Laser Output Parameters

Since the temperature of the active fiber increased with increasing pump power and, accordingly, the cavity length increased, and the HR-FBG was heated, which led to an increase in its period, to ensure single-frequency generation at each pump power, the temperature of the output grating thermostat was optimized (Figure 5a). With temperature optimization, the maximum output power corresponded to a single-frequency operation, while using a nonoptimal setting, lasing in two longitudinal modes was observed (Figure 5b). The optimal pump power was 100 mW, while the output laser power after a Faraday isolator (1.4 dB losses) was 10 mW. With a further increase in the pump power, no increase in the laser output power was observed. We believe that the sharp break in the dependence of the output power on the pump power was due to the negative effect of the higher modes of the active fiber. This was also confirmed by the degradation of the stability of the laser at a pump power of more than 100 mW. We believe that, at a high pump power, the higher modes can obtain a sufficient gain to lead either to parasitic generation in the cavity, which can be formed by Fresnel reflection at the interface between the phosphate and silicate cores, or to an increase in enhanced spontaneous emission. All this leads to a deterioration in stability and a drop in the generation efficiency in the fundamental mode. The pump conversion efficiency was about 10%. The low efficiency was primarily due to the high reflection coefficient of the output grating, in addition to relatively large losses during splicing of active and passive fibers.

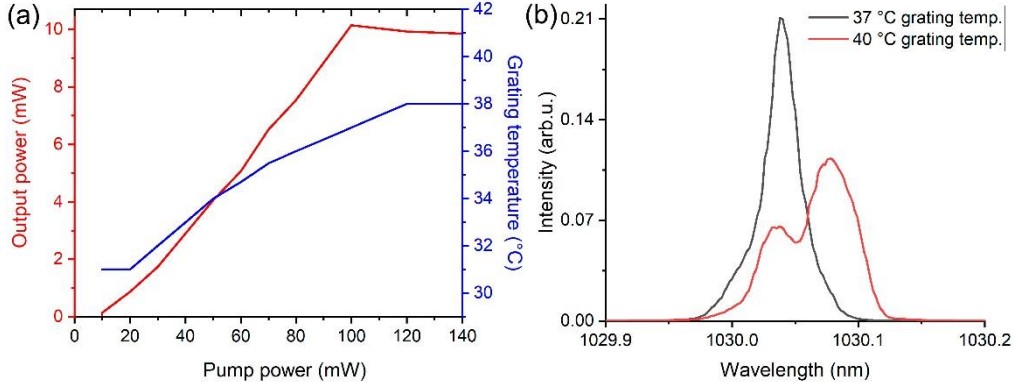

**Figure 5.** (**a**) Laser output power versus pump power (red line). Optimal output grating temperature versus pump power (blue line). (**b**) Laser spectra in the case of optimal (37 °C) and non-optimal (40 °C) output grating temperatures, recorded at a pump power of 100 mW.

To study the polarization, the laser radiation was collimated immediately after leaving the WDM coupler, while the coupler fibers were kept as straight as possible. Polarization was studied using a Glan prism located immediately after the collimator. We measured the polarization extinction (the ratio of the maximum transmission of the Glan prism to the minimum) to be about 30 dB, which indicates that the radiation at the laser output is linearly polarized. The resulting dependence of the polarization extinction on the rotation angle of the Glan prism is shown in Figure 6b.

The signal spectrum (Figure 6a) at the laser output was recorded by an ANDO AQ6317 optical spectrum analyzer. The laser generated radiation at a central wavelength of 1030 nm. The spectrum width was limited by the spectral resolution of the spectrum analyzer (10 pm) and was estimated to be 19 pm. The spectral linewidth was smaller than the cavity mode pitch (25.8 pm), which also confirms the single-frequency operation of the laser. The signal spectrum was stable over time, and the spectral contrast was 65 dB. Analysis of the oscillogram at the output of the Thorlabs DET08C InGaAs photodiode showed no ripples in the laser output signal.

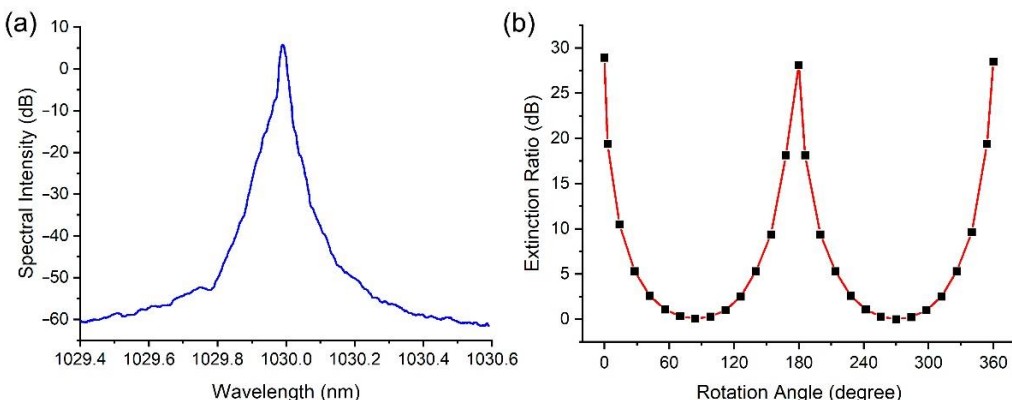

**Figure 6.** (**a**) The spectrum at the output of the developed single-frequency laser. (**b**) Polarization extinction versus angle of rotation of the Glan prism located at the laser output (without Faraday isolator).

We also observed the long-term stability of laser generation using the Thorlabs S145C integrating sphere. Figure 7 shows a three-hour recording of the output power of the developed single frequency laser. The stability of the output power is about 1.5%. We believe that the fluctuations in the output power are mainly due to the fluctuations in the output power of the pump diode, which can be caused by temperature changes.

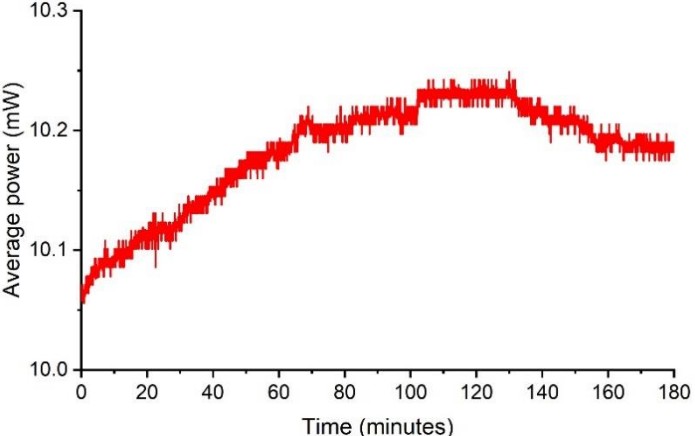

**Figure 7.** Single-frequency laser output power versus time.

## 4. Discussion

In this article, we have shown a single-frequency ytterbium laser built on a composite active fiber, which has a phosphate glass core and a silica cladding. This design eliminates several problems inherent in phosphate fibers: the sensitivity to moisture and the fragility of splices with silica fibers, while providing a high linear gain. However, the current active fiber implementation has the disadvantage of its multimode nature. This did not prevent us from creating a reliable and stable single-frequency source, but the multimode active fiber significantly limits the output power of the laser. The multimode nature of the fiber also makes it impossible to write Bragg gratings directly in the active fiber, so the large intracavity losses caused using external Bragg gratings do not allow high efficiency to be achieved. Thus, the laser we demonstrated does not have any outstanding characteristics. However, the main goal of this work is to show that composite fibers can be good candidates for creating reliable and weather-resistant single-frequency fiber lasers. We believe that the creation of a single-mode composite ytterbium fiber will make it possible to overcome the existing difficulties. The use of a single-mode fiber will allow the recording of Bragg gratings directly in the active fiber, which will significantly reduce the losses in the resonator and also make it possible to avoid instability at high pump powers. We are

currently working on the creation of a composite fiber with a smaller core diameter, which will allow single-mode operation of such a fiber. Reducing the core diameter also requires increasing the concentration of ytterbium ions, since the effective area of the fundamental mode will be smaller. Thus, the creation of a single-mode composite fiber is a rather complicated, but solvable, task. We believe that composite ytterbium fibers can seriously compete with phosphate fibers in the development of fiber lasers with a short cavity, in particular, single-frequency lasers.

## 5. Conclusions

We have demonstrated a single-frequency ytterbium laser based on a composite active fiber, the core of which is made of highly doped phosphate glass and the cladding of silica. The use of such a design makes it possible to provide a high concentration of active ions in the core of the fiber while maintaining all the advantages inherent in silica fibers, such as ease of splicing and lack of degradation due to the influence of the atmosphere. Even though the active fiber is multimodal, the use of Bragg gratings written in a single-mode fiber made it possible to ensure stable operation of such a laser in the single-frequency mode with an output power of 10 mW and a spectral contrast of more than 60 dB. The multimode nature of the active fiber, however, limits the output power of the laser, but we believe that the creation of a single-mode composite fiber will significantly improve the performance of a single-frequency laser built on it.

**Author Contributions:** Conceptualization, O.N.E. and A.V.K.; methodology, O.N.E.; validation, M.Y.K. and O.N.E.; formal analysis, M.Y.K.; investigation, M.Y.K. and A.E.Z.; resources, O.I.M., B.I.G., S.E.S. and B.I.D.; writing—original draft preparation, M.Y.K.; writing—review and editing, O.N.E. and A.V.K.; visualization, M.Y.K.; supervision, A.V.K.; project administration, A.V.K. and S.L.S.; funding acquisition, A.V.K. All authors have read and agreed to the published version of the manuscript.

**Funding:** This research was supported by the Center of Excellence «Center of Photonics» funded by the Ministry of Science and Higher Education of the Russian Federation, contract No. 075-15-2022-316.

**Institutional Review Board Statement:** Not applicable.

**Informed Consent Statement:** Not applicable.

**Data Availability Statement:** Data underlying the results presented in this paper are not publicly available at this time but may be obtained from the authors upon reasonable request.

**Conflicts of Interest:** The authors declare no conflict of interest.

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
