# Peer review of "Narrow-Linewidth Single-Frequency Ytterbium Laser Based on a New Composite Yb3+-Doped Fiber"

_photonics, doi:10.3390/photonics9100760_

Round 1

Reviewer 1 Report

A single-frequency YDFL using composite active fiber was proposed. The main goal of this work is to show that composite fibers can be good candidates for creating reliable and weather-resistant single-frequency fiber lasers. The manuscript was well written. However, some issues should be addressed before it could be accepted.

1.        The author mentioned “by splicing it with standard single-mode fibers, it is possible to avoid excitation of higher modes, especially when working with short sections of the active fiber.” Why higher modes could be avoid?It should be clearly explained.

2.        The author mentioned “lasing was observed in two longitudinal modes which appeared as two peaks in the optical spectrum.” Could the corresponding results be provided?

3.        The author mentioned “while the output laser power after a Faraday isolator (1.4 dB losses), was 10 mW.” However, why the output power in Figure.5 was less than 7 mW?

4.        The linewidth of the laser was measured by analyzing the width of the peaks in Figure 4. More detail should be presented with respect to this method.

5.        The stability of the laser should be quantified within a certain period of observation time.

6.        A performance comparison should be presented with YDFLs proposed by other researchers to demonstrate the advantages and disadvantages of the laser.

7.        Noted that the author mentioned that the fiber laser is weather-resistant. However, the corresponding experiment was not carried. It is suggested that corresponding experiments should be carried out.

Author Response

Reviewer 1

A single-frequency YDFL using composite active fiber was proposed. The main goal of this work is to show that composite fibers can be good candidates for creating reliable and weather-resistant single-frequency fiber lasers. The manuscript was well written. However, some issues should be addressed before it could be accepted.

  1. The author mentioned “by splicing it with standard single-mode fibers, it is possible to avoid excitation of higher modes, especially when working with short sections of the active fiber.” Why higher modes could be avoid? It should be clearly explained.

When a multimode fiber is coaxially excited, the fundamental mode is predominantly excited by a single-mode fiber. Higher modes are also excited, but with less efficiency. Thus, higher modes have large losses in the cavity and, up to a certain pump power, do not take part in the lasing. I have also added some explanations to the article:

“Nevertheless, the single-mode operation of such an optical fiber can be ensured by splicing it with standard single-mode silica fibers. This is possible, since when radiation is coaxially coupled from a single-mode fiber into a multimode one, the fundamental mode is predominantly excited in the latter. Therefore, if the laser cavity is formed by Bragg gratings recorded in a single-mode fiber, single-mode generation is possible in such a laser, despite the multimode nature of the active fiber.”

  1. The author mentioned “lasing was observed in two longitudinal modes which appeared as two peaks in the optical spectrum.” Could the corresponding results be provided?

We added the comparison of two spectra with optimal and non-optimal grating temperature to the article (see Figure 5b).

  1. The author mentioned “while the output laser power after a Faraday isolator (1.4 dB losses), was 10 mW.” However, why the output power in Figure.5 was less than 7 mW?

We agree with this comment, the scale in Figure 5 has not been corrected properly by mistake. Figure 5 has been corrected in the article.

  1. The linewidth of the laser was measured by analyzing the width of the peaks in Figure 4. More detail should be presented with respect to this method.

We have added some explanations to the article describing this method:

“By analyzing the width of the peaks in Figure 4, one can estimate the width of the laser line using a simple formula ..., where ... is the estimated linewidth of the laser, FSR is the free spectral range of the interferometer, ... is the duration of the peak in Figure 4, and ... is the time between the pulses, which corresponds to the achievement of the FSR of the interferometer during scanning (1.5 GHz).”

However, we would like to note that this method gives an upper estimate of the spectral width and is limited by the resolution of the scanning interferometer (the resonator mode width).

  1. The stability of the laser should be quantified within a certain period of observation time.

We observed the laser for several hours and did not notice changes in its average power by more than a few percent, which is most likely due to the instability of the pump radiation. I have added a three-hour recording of the laser output to the manuscript along with the appropriate comments (see Figure 7).

  1. A performance comparison should be presented with YDFLs proposed by other researchers to demonstrate the advantages and disadvantages of the laser.

As mentioned in the article, the performance of our laser is not outstanding. As can be seen from Refs. [12] and [14] of the original manuscript, phosphate glass fiber lasers can reach hundreds of milliwatts of output power with the same spectral contrast and linewidth. The main advantage of our laser is the weather resistant design of the active fiber, which we believe will help to improve characteristics of the short cavity single frequency fiber laser. However, it is just a first attempt of making such a laser at 1030 nm wavelength.

  1. Noted that the author mentioned that the fiber laser is weather-resistant. However, the corresponding experiment was not carried. It is suggested that corresponding experiments should be carried out.

As we mentioned in the article, the low strength of splicing of phosphate and quartz fibers, as well as the deterioration of the strength of phosphate fibers under the influence of the atmosphere, is a well-known problem caused by the properties of phosphate glass. In our fiber, a small core of phosphate (moisture sensitive) glass is surrounded by a thick cladding of standard silicate glass. Thus, when spliced with standard fibers, our fiber has the same weather resistance parameters as silicate fibers commonly used throughout.

Reviewer 2 Report

In this paper, a new composite structure single frequency fiber laser is reported. This new composite structure can alleviate the need to ensure higher splicing strength of optical fibers and increase the resistance to atmospheric moisture. Although the final results are not ideal, the work is very interesting. Therefore, this manuscript is worth publishing in photonics. However, the author needs to address my concerns as follows.

1.      The English expression may be modified appropriately, and the references should be checked carefully.

2.      The formats of Fig. 4 and Fig. 5 remain unified, and Fig. 2 needs to be modified as a whole.

3.      The theoretical part is too rough and needs some optimization.

4.      Is this composite optical fiber produced by your team?

5.      Now the experiment is not good, if your team has any ways to improve it, you can put it in the paper.

Author Response

Reviewer 2

In this paper, a new composite structure single frequency fiber laser is reported. This new composite structure can alleviate the need to ensure higher splicing strength of optical fibers and increase the resistance to atmospheric moisture. Although the final results are not ideal, the work is very interesting. Therefore, this manuscript is worth publishing in photonics. However, the author needs to address my concerns as follows.

  1. The English expression may be modified appropriately, and the references should be checked carefully.

I have checked all the references. Unfortunately, as a result of an error in some places, instead of reference [19], there was reference [17]. Now this error has been fixed.

  1. The formats of Fig. 4 and Fig. 5 remain unified, and Fig. 2 needs to be modified as a whole.

I don't understand this comment because Figure 2 is a schematic of the experimental setup and Figures 4 and 5 are graphs. They will always be different in style, because they have different purposes.

  1. The theoretical part is too rough and needs some optimization.

This article is mainly devoted to the creation and experimental study of a single-frequency laser. We have made corrections in some places, hopefully now it sounds better.

  1. Is this composite optical fiber produced by your team?

Yes, this fiber was produced by our team. You may have been misled by Ref. [17] in the active fiber description of the previous version. This reference was posted by mistake and has now been replaced by Ref. [19].

“19. Egorova, O.N.; Semenov, S.L.; Medvedkov O.I.; Astapovich, M.S.; Okhrimchuk, A.G.; Galagan, B. I.; Denker, B.I.; Sverchkov, S.E.; Dianov, E.M. High-beam quality, high-efficiency laser based on a fiber with heavily Yb3+-doped phosphate core and silica cladding. Optics Letters 2015, 40, 3762–3765. doi: 10.1364/OL.40.003762.”

  1. Now the experiment is not good, if your team has any ways to improve it, you can put it in the paper.

Yes, as mentioned in the article, we plan to manufacture a single-mode composite fiber to solve the main problem that prevents high output powers from being achieved - the multimode nature of the active fiber. We've updated the discussion section of the manuscript concerning this point.

Reviewer 3 Report

In the manuscript ' Narrow-linewidth single-frequency ytterbium laser based on a new composite Yb3+-doped fiber, ' the authors Koptev et al. reported their research on the single-frequency laser using composite fiber. Personally, I do not think the novelty of this manuscript is attractive enough to the readers based on current version. Therefore, the authors must answer the following questions before this manuscript can be accept.

1. There is no sufficient discussion on the limited performance of the single-frequency laser and the author must give more discussion on the reason and the possible technique map to improve it. As we know, if the performance is not outstanding, the novelty of this manuscript is limited.

2. The authors just give several sentences describing the polarization of the laser which is not enough. The figure showing the relation between the contrast and the angle of half wave-plate should be given. 

Author Response

Reviewer 3

In the manuscript ' Narrow-linewidth single-frequency ytterbium laser based on a new composite Yb3+-doped fiber, ' the authors Koptev et al. reported their research on the single-frequency laser using composite fiber. Personally, I do not think the novelty of this manuscript is attractive enough to the readers based on current version. Therefore, the authors must answer the following questions before this manuscript can be accept.

  1. There is no sufficient discussion on the limited performance of the single-frequency laser and the author must give more discussion on the reason and the possible technique map to improve it. As we know, if the performance is not outstanding, the novelty of this manuscript is limited.

I've added some explanations about the limitations of the current laser:

“We believe that, at a high pump power, the higher modes can obtain a sufficient gain to lead either to parasitic generation in the cavity, which can be formed by Fresnel reflection at the interface between the phosphate and silicate cores, or to an increase in enhanced spontaneous emission. All this leads to a deterioration in stability and a drop in the generation efficiency in the fundamental mode.”

 I also updated the discussion section of the manuscript, providing our way to overcome current limitations.

  1. The authors just give several sentences describing the polarization of the laser which is not enough. The figure showing the relation between the contrast and the angle of half wave-plate should be given.

We added a new figure of the dependence of the polarization extinction on the angle of rotation of the Glan prism to the article (see Figure 6b).

Reviewer 4 Report

Dear authors,

It would be more informative if you measured laser linewidth by self-heterodyne technique to obtain more presicion value.

And could you please to write a bit more about the possibility of creation of a single-mode composite ytterbium fiber? Have you already made any attempts in this direction?

Author Response

Reviewer 4

It would be more informative if you measured laser linewidth by self-heterodyne technique to obtain more presicion value.

Certainly, measuring linewidth by the self-heterodyne technique is more precise, but at the moment we cannot make such measurements. Nevertheless, the measurements carried out with the Fabry-Perot interferometer allow us to speak of a rather narrow spectral line of the laser.

And could you please to write a bit more about the possibility of creation of a single-mode composite ytterbium fiber? Have you already made any attempts in this direction?

Yes, we are currently working on the creation of a single-mode ytterbium composite fiber. The fabrication of such a fiber requires a reduction in the core diameter and, at the same time, an increase in the concentration of ytterbium ions. We have added information about this to the manuscript (see the discussion section).

Round 2

Reviewer 1 Report

Can be accepted.